# Basicranial Modular Organization. A Study in the Araucanian Horse of Colombia

**DOI:** 10.3390/vetsci10040255

**Published:** 2023-03-28

**Authors:** Arcesio Salamanca-Carreño, Pere M. Parés-Casanova, René Alejandro Crosby-Granados, Mauricio Vélez-Terranova, Jannet Bentez-Molano

**Affiliations:** 1Facultad de Medicina Veterinaria y Zootecnia, Universidad Cooperativa de Colombia, Villavicencio 500001, Colombia; 2Institució Catalana d’Història Natural, 08001 Barcelona, Spain; 3Facultad de Ciencias Agropecuarias, Universidad Nacional de Colombia, Palmira 763531, Colombia

**Keywords:** arauca, facial skeleton, neurocranium, splanchnocranium, viscerocranium

## Abstract

**Simple Summary:**

The comparison of anatomical characters is scarcely investigated in domestic animal breeds, and the racial classification has been based on morphological descriptions of the cranial region as a unit. The aim was to study the basicranial organization of the neurocranium and splanchnocranium modules in a sample of 31 skulls of adult Araucanian horses using 2D morphometric geometric techniques. Thirty-one reference points were used. The RV coefficient (the multivariate analog of a correlation) was estimated to analyze the independence of these two parts, as well as their morphological integration, using a two-block least squares analysis. The study results confirm the modular development of the neurocranium and the splanchnocranium, the former being more stable than the latter as well as low morphological integration between the two. The development between both parties is structured in a modular way but allows relative independence. Now it would be interesting for future studies to add muscles (those that connect the cranial parts, but also the cervical), the hyoid apparatus, and the ossicles of the internal ear and the jaw and analyze if they behave as integrated modules between them.

**Abstract:**

The skull is divided into neurocranium and splanchnocranium, and its variation allows ecomorphological studies to learn about possible evolutionary and adaptive characteristics. The basicranial organization of the neurocranium and splanchnocranium modules was studied in a sample of 31 skulls from adult Araucanian horses by means of 2D geometric morphometric techniques. The neurocranium and splanchnocranium modules on the ventral aspect were analyzed separately using a set of 31 landmarks. The RV coefficient (the multivariate analog of a correlation) was estimated to analyze the independence of these two parts, as well as their morphological integration, using a two-block analysis of least squares. The study results confirm the modular development of the neurocranium and the splanchnocranium, the former being more stable than the latter as well as low morphological integration between the two. The development between both parties is structured in a modular way but allows relative independence. Now it would be interesting for future studies to add muscles (those that connect the cranial parts, but also the cervical), the hyoid apparatus, and the ossicles of the internal ear and the jaw and analyze if they behave as integrated modules between them. Since this research has been conducted at the subspecific breed level, it could be plausible that in other breeds, this integrative development was different.

## 1. Introduction

The skull is a highly complex bone structure made up of a varied set of bones [1]. These bones, from a morphological perspective, show differential growth and development [2], largely the way in which these processes occur that determines the cranial morphological variation [2]. The skull is made up of two easily distinguishable anatomical segments: the splanchnocranium (viscerocranium or facial complex) and the neurocranium [3,4]. This second segment functions to protect the brain and can be divided into the base and the cranial vault [4]. As these complex structures are influenced by differential growth and development, it is therefore hypothesized that ecomorphological diversity is likely due to the specialization of such structures to a new environment [5].

Morphological and osteometric studies provide valuable information on the characteristics and potential use of some animal species while providing evidence of low growth rates generating morphological divergence [6,7]. This had the potential to impact morphological modulation and influence those correlating structures involved in overall function. Such modular organization and segmentation are well-established in mammals [8,9].

Modularity refers to the relative independence of certain structures, while morphological integration refers to how these units covary with each other, and in geometric morphometry, “modules” are defined as units within which there is a high integration among them [8]. This integration manifests itself as a marked covariation of the components or parts within that same module [8] and is possible since the components of the skull evolve, develop, and function in a joint and coordinated manner [10]. By combining multivariate statistical methods and morphometry, the phenomenon of modularity and morphological integration of skulls can be studied [11].

Horses have played a significant role in shaping cultural and ecological systems in both indigenous and colonial societies [12]. The comparison of anatomical characters is a part that has been scarcely investigated in domestic animal breeds, and traditionally, racial classification has been based on morphological descriptions of the cranial region as a unit. In the present work, a sample of skulls from Araucana breed horses is analyzed, testing the hypothesis of 2 modules in the basicranium: splanchnocranium part and neurocranium part. The Araucanian horse constitutes an equine population typical of the ecosystem of the Araucanian plains in NE Colombia. Of great rusticity, it has hardly been studied in its ecological environment until recently, when investigations have been initiated aimed at its morphological description, functional study, and genetic management [13,14,15]. The Araucanian horse represents a cultural and genetic heritage adapted to the environment of the flooded savanna; therefore, a morphological study might reveal its evolutionary traits of adaptation to the environment.

## 2. Materials and Methods

### 2.1. Sampling

The specimens examined in this study came from the mastological collection of the Faculty of Veterinary Medicine and Zootechnics, Universidad Cooperativa de Colombia. Colombia. The samples included a total of 31 skulls from adult animals, mostly males (although most came from entire males, sex was not considered for the analysis). The skulls had been collected in the Araucanian plains ecosystem, and all had at least an erupted M2 and, therefore, were older than 2 years [16] and without osteological manifestations of problems. No request for ethical approval was considered since skulls already collected for research and exhibition purposes were used.

### 2.2. Primary Data Collection

Each skull was placed on the floor in a horizontal position, and a photograph was taken in the ventral plane using a Nikon P530 42x optical camera. Each skull was previously placed in the center of the optical field, with its ventral face oriented to the camera. To eliminate distortions of the relative positions of the angles due to the parallax (depth) effect [17], the camera was placed at a sufficient distance to ensure that the skull occupied only a part of the visual field, free of distortions. Failure to consider the coplanarity criterion could lead to a misinterpretation of the results since there may be important variations in the shape that are not appreciable in the parallel plane to the photograph [18]. A millimeter standard was included in each shot. The images were stored in a JPEG extension and later transferred to the computer. Digital photographic documents are protected by the second author. Seventeen anatomical landmarks were located in each specimen, 5 sagittal and 6 bilateral pairs (Figure 1), partially based on previous studies [19]. All craniometric points (landmarks) considered corresponded to points defined from homologous and repeatable anatomical referents and were considered sufficient to reflect the morphology on the ventral side of the neurocranium and splanchnocranium. These landmarks were located in each specimen using the TpsDig version 1.40 program [20].

### 2.3. Processing of the Primary Data

The selected landmarks generated a matrix of coordinates (X and Y) that represent the geometric configurations of the skull per specimen. The matrix of coordinate configurations was subjected to a Generalized Procrustes Analysis [21], where the variation associated with the effects of position, orientation and scale was eliminated by means of a Procrustes overlay [22], using the estimation of least squares [23,24]. In general terms, this analysis consists of three iterative steps: (1) each configuration is centered at the origin of the coordinate system and then adjusted to a common size unit; (2) the scaled configurations translate one over the other overlapping, in such a way that their centers of gravity (centroids) coincide; and (3) they are rotated until the distances between each milestone and an average configuration are minimized, using the mathematical criterion of least squares [18,25,26]. The variation of the shape of the specimens with the tangent space was analyzed with the TpsSmall program version 1.33 [27]. The correlation of the shape of the specimens with the tangent space -in which the configurations are projected orthogonally [18] was analyzed with the TpsDig version 1.40 program [20]. A principal component analysis was carried out to detect the most discriminating variables [28], and finally, the regression study was carried out between the centroid size, logarithmically transformed, as an independent variable, as an expression of the size of the specimen and the Procrustes x and y coordinates of each data point reference as dependent variables. A Principal Components Analysis was performed from the regression residuals to detect the most discriminating variables. When considering the shape, we worked with the symmetrical component since bilateral asymmetry processes cannot be ruled out [14], which could reflect different results depending on the side studied.

To test the modularity hypothesis, “the partition of the homologous reference configuration was specified into two subsets that correspond to two hypothetical cranial modules” [29] based on differential embryonic origins: splanchnocranium (landmarks 1–3 and 6–11) and neurocranium (landmarks 4 and 5 and 12–17; Figure 2), also from the regression residuals. Although it is spatially related to facial structures, the orbital region (landmarks 12–13 and 16–17) was considered to belong to the neurocranium. Covariance between modules was compared against several possible alternative partitions with the same number of homologous milestones as the hypothesized modules [29] by calculating the RV coefficient [8]. “This coefficient can be considered a multivariate analog of a, Escoufier correlation” [30], “and it was calculated between the two hypothetical modules and between the set of alternative partitions, generating a distribution of values” [29,31].

The partial least squares (PLS) method is used to study patterns of covariation between two or more sets of variables. “PLS consists of finding correlated pairs of linear combinations (singular vectors) between two sets (or blocks) of variables, which account for the greatest possible covariation between the two blocks of original variables with the aim of trying to maximize the representation in few dimensions of the covariance structure between the sets of variables (or blocks)” [29].

All analyzes were carried out using the MorphoJ package version 1.06d [32].

## 3. Results

### 3.1. Preliminary Analysis

Preliminary data analysis indicated an excellent correlation between tangent and shape space, with the uncentered correlation being between the regression of tangent space, Y, and the Procrustes distance (correlation value 0.999629). This confirmed that the samples could be analyzed by geometric morphometry.

### 3.2. Allometry

In the regression, it was reflected that 21.7% of the variation in the shape was explained by the size (*p* < 0.0001; 10,000 rounds of permutations). For the splanchnocranium, this variation was 14.9% (*p* = 0.0023), while for the neurocranium, it rose to 54.5% (*p* < 0.0001).

### 3.3. Principal Component Analysis

In the Principal Component Analysis (PCA), the first three principal components explained 66.4% of the total variation observed (PC1 + PC2 + PC3 = 39.38% + 16.95% + 10.09%). The most discriminating coordinates are reflected in Table 1 and correspond to lengths.

### 3.4. Modularity and Integration

The result obtained shows a perfect correspondence in each module, which supports the “modular division based on differential origins of development” (neurocranium and splanchnocranium). “The distribution of the RV coefficients shows that the a priori hypothesis” is at the extreme left of the distribution curve, being the partition with the lowest RV coefficient (RV = 0.337080; percentage of alternative partitions with an RV value lower than that of the hypothetical modules: 2.91%; Figure 3). Of a total of 291 alternative partitions for “which the RV coefficient was calculated,” none presented an RV coefficient lower than that previously hypothesized. This result is, therefore, “good evidence to support the modular organization by differential origins” of development.

### 3.5. Morphological Integration

After verifying that splanchnocranium and neurocranium behave in modules, we proceeded to study the covariation between both modules to calculate the level of morphological integration between them. The RV coefficient was used here as a general measure for the strength of the association between the blocks. As previously explained, this coefficient can be interpreted as a multivariate analog of a correlation coefficient between two variables [30]. The correlation between the PLS1 scores was 0.695 (*p* = 0.404, after 1000 rounds of permutation). This means that there is no significant morphological integration between neurocranium and splanchnocranium. Figure 4 shows the covariation pattern of the first pair of singular axes, visually demonstrating the low covariation between them.

The first two pairs of singular axes accounted for 90.6% of the total covariance of the sample (PLS1 + PLS2 = 79.18% + 12.47%), while the second two pairs explained only 8.22% (PLS3 + PLS4 = 5.85% + 2.36%) (Table 2).

The covariation between the modules for the PLS1 can be described as a slight lateral expansion in the zygomatic and parasagittal portion of the occipital condyles, while the visceral portion presents a rostrocaudal shortening (Figure 5). The neurocranium presented covariation patterns with lower values than in the splanchnocranium.

## 4. Discussion

The result obtained shows a perfect correspondence in each module, which supports the modular division based on development differential origins (neurocranium and splanchnocranium) [33]. The covariation in splanchnocranium and neurocranium, on the other hand, would depend to a great extent on the change in size that affects both regions separately [34].

In this study, it can be stated that there is little integration between the components of the equine skull. That is, there is little integration between the splanchnocranium and the neurocranium, portions of the skull with different embryonic origins. The neurocranium appears to be more stable than the splanchnocranium, probably influenced by the brain, which constitutes its main functional matrix [11]. In other words, most of the shape characteristics of the neurocranium would be established early, probably in response to the influence during the prenatal stage of the organs it contains. In the condylar region, the greatest covariation observed could be due to the influence of muscle loads that have their insertion at the caudal level of the skull. The lesser covariation of the traits of the zygomatic region, on the other side, would be established earlier and would later present minor modifications [35]. The splanchnocranium would be evidenced as a more dynamic module. The strong integration at the level of the splanchnocranium could also be related to functional factors linked to food [11].

Although the functional and developmental dependencies of the head cannot be separated, the bimodular structure at the ventral level of the skull would allow a certain evolutionary independence that the change in one of these modules would affect the other. These structures would vary in an uncoordinated manner due to allometry, which is a key integrating factor, especially due to the prolonged period of facial growth [35]. Since different functions affect the splanchnocranium and neurocranium differently, it is reinforced that they would share a different evolutionary history. Therefore, at the training level of the different horse breeds, both modules would present separate changes: the selection for a character (for example, the shortening of the face in some breeds) would not necessarily imply a change at the general cranial level. This coincides with what was exposed decades ago regarding the change in the facial index (facial length/cranial length) for equine evolution [36] and that we could extend to the formation of breeds for this species.

From an embryological point of view, this relatively independent development of the two modules makes great sense: the neurocranium has two origins: one dermal (the vault), reminiscent of ancestral exoskeletons, and the other cartilaginous (the base), known as the chondrocranium [4]. The splanchnocranium develops in turn from the neural crest. Each of these skull modules grows independently and presents allometry differently. This is because each module receives different stimuli for its development: the splanchnocranium receives stimuli from the facial organs, while the neurocranium receives internal stimuli from the brain and external stimuli from the neck and chewing muscles; in addition, it develops more slowly as its ossification is endochondral [4,11].

## 5. Conclusions

This study confirms the modular development of the neurocranium and the splanchnocranium, the former being more stable than the latter (possibly due to the brain, which constitutes its main functional matrix), as well as low morphological integration between the two. The development between both parties is structured in a modular way but allows relative independence. For this study, we have focused only on the bony parts of the skull. It would now be interesting, for future studies, to add to this type of study muscles (those that connect the cranial parts, but also the cervical ones), the hyoid apparatus, the ossicles of the inner ear, and the mandible, and analyze whether they behave as integrated modules among them.

## Figures and Tables

**Figure 1 vetsci-10-00255-f001:**
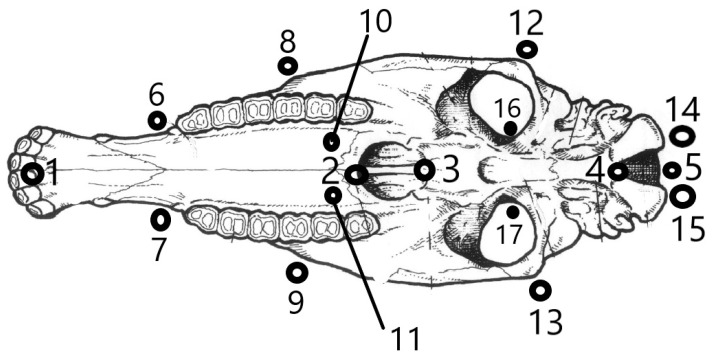
Ventral face of an equine skull showing the landmarks selected for study in the Araucanian horse. Seventeen anatomical landmarks were in each specimen, 5 sagittal and 6 bilateral pairs. Landmarks 1–3 and 6–11 were considered to determine the modulus of the splanchnocranium; landmarks 4 and 5 and 12–17 for the neurocranium module.

**Figure 2 vetsci-10-00255-f002:**
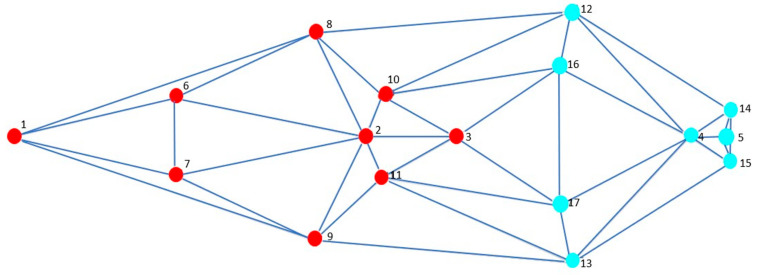
Hypothetical modules considered: splanchnocranium (landmarks 1 to 3, and 6 to 11; red points) and neurocranium (landmarks 4 and 5, and 12 to 17; blue points).

**Figure 3 vetsci-10-00255-f003:**
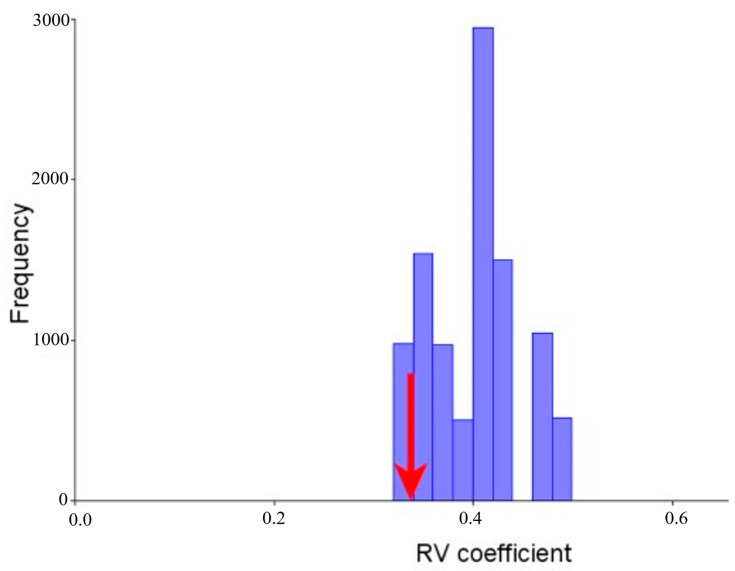
Histogram of the distribution of 291 RV coefficients generated by permutation. The arrow shows that our hypothesis is at the extreme left of the distribution curve, being the partition with the lowest RV coefficient (RV = 0.337080; percentage of alternative partitions with an RV value lower than that of the hypothetical modules: 2.91%; 1000 rounds of permutation).

**Figure 4 vetsci-10-00255-f004:**
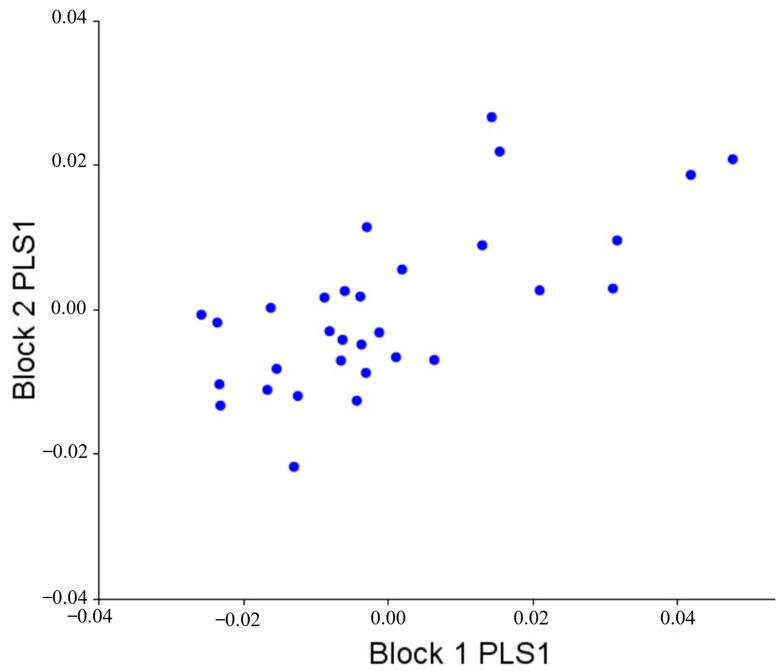
Graph of the PLS1 scores (78.19% of the total variation observed) for the first pair of singular axes of the two modules analyzed: splanchnocranium (Block 1) and neurocranium (Block 2). A low morphological integration between splanchnocranium and neurocranium is observed (*p* = 0.404, after 1000 rounds of permutation).

**Figure 5 vetsci-10-00255-f005:**
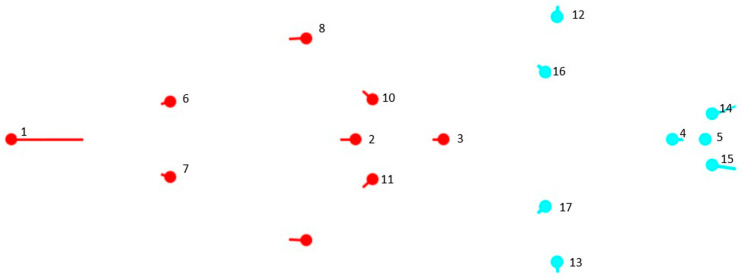
Covariation between modules for module PLS1. There is a slight lateral expansion in the zygomatic and parasagittal portion of the occipital condyles and caudal face shortening in the visceral portion.

**Table 1 vetsci-10-00255-t001:** Most discriminating coordinates in the Principal Components Analysis for the first three components (PC1 + CP2 + CP3 = 39.38% + 16.95% + 10.09%). The most discriminative coordinates (>[0.3]) are highlighted in bold.

	PC1	PC2	PC3
**x1**	**−0.7767**	0.1055	0.1788
y1	0.0000	0.0000	0.0000
x2	0.1595	−0.1770	0.1973
y2	0.0000	0.0000	0.0000
x3	0.1258	−0.2369	−0.1501
y3	0.0000	0.0000	0.0000
x4	−0.1203	−0.2064	0.0537
y4	0.0000	0.0000	0.0000
x5	0.0039	0.2589	−0.0458
y5	0.0000	0.0000	0.0000
x6	0.1263	0.1706	−0.2882
y6	0.0356	0.0436	−0.0970
x7	0.1263	0.1706	−0.2882
y7	−0.0356	−0.0436	0.0970
x8	0.1853	0.0183	−0.1422
y8	−0.0258	0.1779	0.1118
x9	0.1853	0.0183	−0.1422
y9	0.0258	−0.1779	−0.1118
**x10**	0.1349	**−0.3320**	**0.3613**
y10	−0.0653	0.0173	0.0998
**x11**	0.1349	**−0.3320**	**0.3613**
y11	0.0653	−0.0173	−0.0998
**x12**	0.0326	**0.3179**	0.1407
y12	−0.0735	0.1390	0.1781
**x13**	0.0326	**0.3179**	0.1407
y13	0.0735	−0.1390	−0.1781
**x14**	−0.2708	−0.2414	**−0.3096**
y14	−0.0408	0.0772	−0.1520
**x15**	−0.2708	−0.2414	**−0.3096**
y15	0.0408	−0.0772	0.1520
x16	0.0956	0.1945	0.1211
y16	−0.0719	−0.0013	−0.0030
x17	0.0956	0.1945	0.1211
y17	0.0719	0.0013	0.0030

**Table 2 vetsci-10-00255-t002:** Single values and paired correlations for the PLS scores between splanchnocranium and neurocranium.

	Single Value	*p*-Value (Perm.)	% Total Covariación	Correlation	*p*-Value (Perm.)
PLS1	0.00015	0.008	78.198	0.69529	0.404
PLS2	5.98 × 10^−5^	0.148	12.477	0.55855	0.296
PLS3	4.1 × 10^−5^	0.016	5.858	0.71101	0.012
PLS4	2.61 × 10^−5^	0.060	2.366	0.61453	0.020
PLS5	1.37 × 10^−5^	0.556	0.650	0.44533	0.188
PLS6	9.85 × 10^−5^	0.256	0.338	0.46873	0.060
PLS7	0.000005	0.464	0.087	0.40127	0.012
PLS8	2.71 × 10^−6^	0.280	0.026	0.09417	0.536

## Data Availability

Data are available upon reasonable request to the second author.

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
