# Peer review of "Basicranial Modular Organization. A Study in the Araucanian Horse of Colombia"

_vetsci, 2023, doi:10.3390/vetsci10040255_

Round 1

Reviewer 1 Report

Basicranial modular organization. A study in the Araucanian 2 horse of Colombia

Abstract

First sentence is misplaced here. There are statements not established in the manuscript and have been interpreted here. Consider your conclusion at the end of the discussion and then re-address the abstract.

Introduction

Too short and the English grammar requires correction.

The authors need to expand on what is ‘modular organization’; its development inutero and connection to the axial skeleton. What is its relevance to Veterinary Science?   

Materials and Methods

2.1 The authors represent the breed as – working with cattle and here no worked animals are determined. The sample describes ‘mostly male’ from adult animals without derivation of sex, surely the canine tushes would provide evidence of male numbers verse female numbers. At an age of 2, the animal is not adult so sexual dimorphism might be an issue in an immature skull. The last paragraph in this section is irrelevant here and if reworded correctly, would be a better fit at the end of the introduction.

2.2 How did you prep the skull prior to photography – orientation – rostral left or right per animal? How did you determine the landmarks if only some were partially based on previous studies? Your considered sufficient might not correlate to another scientist’s. You need to be more specific. Landmarks must be anatomical described for subsequent research repeatability, this can be achieves by labelling Figure 1 more precisely. 

2.3 Grammar. In Figure 2 the landmarks would be better understood if they overlayed the ventral skull with a transparency factor of 3 or similar. Also, you need to describe the two colours and their relevance. The numbers could also be better defined for clarity. Figures of the rotation described would be beneficial in understanding the terminology, such as ‘tangent space’ .

Results

Grammar.

3.1 Here the results would be better supported by the diagram recommended above.

3.4 – 3.5 This is a results section not discussion. All methods should have been outlined in Materials and Methods.

Discussion

Grammar. There is a presumption that the reader has a firm grasp of anatomical structures without establishing the structural components in the literature beforehand. The discussion brings forward knowledge that should have been previously established in the Introduction, and Material and Methods, e.g.,  embryology, musculature and organs.   

Author Response

First reviewer’s responses

Dear reviewer

The authors appreciate the insightful comments.

We attach all corrections and answers

Comment

Abstract

First sentence is misplaced here. There are statements not established in the manuscript and have been interpreted here. Consider your conclusion at the end of the discussion and then re-address the abstract.

Response

Corrected

Comment

Introduction

Too short and the English grammar requires correction.

The authors need to expand on what is ‘modular organization’; its development inutero and connection to the axial skeleton. What is its relevance to Veterinary Science?  

Response

The modular organization refers to the way the different features of an organism correlate

The Araucanian horse represents a cultural and genetic heritage adapted to environmental conditions of the flood savanna; Therefore, its study is justified because it is necessary to know its adaptative and evolutionary traits

Corrected and added in the text

Comment

Materials and Methods

2.1 The authors represent the breed as – working with cattle and here no worked animals are determined. The sample describes ‘mostly male’ from adult animals without derivation of sex, surely the canine tushes would provide evidence of male numbers verse female numbers. At an age of 2, the animal is not adult so sexual dimorphism might be an issue in an immature skull. The last paragraph in this section is irrelevant here and if reworded correctly, would be a better fit at the end of the introduction.

Response

Corrected and inserted in manuscript 

2.2 How did you prep the skull prior to photography – orientation – rostral left or right per animal? How did you determine the landmarks if only some were partially based on previous studies? Your considered sufficient might not correlate to another scientist’s. You need to be more specific. Landmarks must be anatomical described for subsequent research repeatability, this can be achieves by labelling Figure 1 more precisely. 

Response

This explanation is in 2.1. The skulls were prepared in the Mastology Laboratory of the Cooperative University of Colombia. The skulls were placed on the floor and a photograph was taken in the ventral plane.

  Was corrected in the text

2.3 Grammar. In Figure 2 the landmarks would be better understood if they overlayed the ventral skull with a transparency factor of 3 or similar. Also, you need to describe the two colours and their relevance. The numbers could also be better defined for clarity. Figures of the rotation described would be beneficial in understanding the terminology, such as ‘tangent space’ .

Response

Corrected in the text

Comment

Results

Grammar.

3.1 Here the results would be better supported by the diagram recommended above.

3.4 – 3.5 This is a results section not discussion. All methods should have been outlined in Materials and Methods.

Response

The Results section is separated from Discussion in the manuscript.

Comment

Discussion

Grammar. There is a presumption that the reader has a firm grasp of anatomical structures without establishing the structural components in the literature beforehand. The discussion brings forward knowledge that should have been previously established in the Introduction, and Material and Methods, e.g.,  embryology, musculature and organs.   

Response

Comments were corrected in the text

English was corrected

Reviewer 2 Report

Dear Authors, please correct the marked sentences!

Author Response

Dear reviewer

The authors appreciate the insightful comments.
Suggested corrections were made and included in the text

Round 2

Reviewer 1 Report

INTRODUCTION

From line 45 – rephrase – suggest -

The skull is made up of two easily distinguishable anatomical segments: the splanchnocranium (viscerocranium or facial complex), and the neurocranium [3,4]. This second segment functions to protect the brain and can be divided into the base and the cranial vault [4]. As these complex structures are influenced by differential growth and development, it is therefore hypothesised that ecomorphological diversity is likely due to the specialization of such structures to a new environment [5].

Morphological and osteometric studies provide valuable information on the characteristics and potential use of some animal species while providing evidence of low growth rates generating morphological divergence [6,7]. This has the potential to impact morphological modulation and influence those correlating structures involved in overall function. Such modular organisation and segmentation is well established in mammals (8,9).       

From line 68 – rephrase – suggest –

Of great rusticity, it has hardly been studied in its ecological environment, until recently, when investigations have been initiated aimed at its morphological description, functionality, and genetic management [13,14,15]. The Araucanian horse represents a cultural and genetic heritage adapted to environmental conditions of the flood savanna; therefore, a morphological study might reveal its evolutionary traits of adaptation to the environment.

MATERIALS AND METHODS

Figure 2 – the numbers are too hard to read.

RESULTS

Figure 5 - the numbers are too hard to read.

DISCUSSION

First sentence better suited in the INTRODUCTION.

Author Response

First reviewer’s responses Round2

Dear reviewer

The authors appreciate the insightful comments.

We attach all corrections and answers

Comment

NTRODUCTION

From line 45 – rephrase – suggest -

The skull is made up of two easily distinguishable anatomical segments: the splanchnocranium (viscerocranium or facial complex), and the neurocranium [3,4]. This second segment functions to protect the brain and can be divided into the base and the cranial vault [4]. As these complex structures are influenced by differential growth and development, it is therefore hypothesised that ecomorphological diversity is likely due to the specialization of such structures to a new environment [5].

Response

Rephrased and included in the text.

Morphological and osteometric studies provide valuable information on the characteristics and potential use of some animal species while providing evidence of low growth rates generating morphological divergence [6,7]. This has the potential to impact morphological modulation and influence those correlating structures involved in overall function. Such modular organisation and segmentation is well established in mammals (8,9).      

Response

Rephrased and included in the text.

From line 68 – rephrase – suggest –

Of great rusticity, it has hardly been studied in its ecological environment, until recently, when investigations have been initiated aimed at its morphological description, functionality, and genetic management [13,14,15]. The Araucanian horse represents a cultural and genetic heritage adapted to environmental conditions of the flood savanna; therefore, a morphological study might reveal its evolutionary traits of adaptation to the environment.

Response

Rephrased and included in the text.

Comment

MATERIALS AND METHODS

Figure 2 – the numbers are too hard to read.

Response

Figure 2 was corrected and included in the text.

Comment

RESULTS

Figure 5 - the numbers are too hard to read.

Response

Figure 5 was corrected and included in the text.

Comment

DISCUSSION

First sentence better suited in the INTRODUCTION.

Response

Corrected. The sentence was included of the text introduction.

English was corrected
